# Vertical Ridge Augmentation of Fibula Flap in Mandibular Reconstruction: A Comparison between Vertical Distraction, Double-Barrel Flap and Iliac Crest Graft

**DOI:** 10.3390/jcm10010101

**Published:** 2020-12-30

**Authors:** Carlos Navarro Cuéllar, Santiago Ochandiano Caicoya, Ignacio Navarro Cuéllar, Salvador Valladares Pérez, Rodrigo Fariña Sirandoni, Raúl Antúnez-Conde, Alberto Díez Montiel, Arturo Sánchez Pérez, Ana María López López, Carlos Navarro Vila, José Ignacio Salmerón Escobar

**Affiliations:** 1Maxillofacial Surgery Department at Hospital General Universitario Gregorio Marañón, C/Doctor Esquerdo 46, 28007 Madrid, Spain; sochandiano@hotmail.com (S.O.C.); nnavcu@hotmail.com (I.N.C.); antunezconde_92@hotmail.com (R.A.-C.); diezmontiel@gmail.com (A.D.M.); anitalopez@gmail.com (A.M.L.L.); hazamachado2@gmail.com (C.N.V.); jisalmeron@telefonica.net (J.I.S.E.); 2Maxillofacial Surgery Department at Hospital Clínico Metropolitano El Carmen—Maipú, 9250000 Santiago, Chile; salvadorvalladares@gmail.com; 3Maxillofacial Surgery Department at Hospital del Salvador, 7500000 Santiago, Chile; rofari@gmail.com; 4Faculty of Medicine, Murcia University, 30008 Murcia, Spain; arturosa@um.es

**Keywords:** fibula flap, vertical augmentation, double-barrel flap, vertical distraction, iliac crest graft

## Abstract

Double-barrel flap, vertical distraction and iliac crest graft are used to reconstruct the vertical height of the fibula. Twenty-four patients with fibula flap were reconstructed comparing these techniques (eight patients in each group) in terms of height of bone, bone resorption, implant success rate and the effects of radiotherapy. The increase in vertical bone with vertical distraction, double-barrel flap and iliac crest was 12.5 ± 0.78 mm, 18.5 ± 0.5 mm, and 17.75 ± 0.6 mm, (*p* < 0.001). The perimplant bone resorption was 2.31 ± 0.12 mm, 1.23 ± 0.09 mm and 1.43 ± 0.042 mm (*p* < 0.001), respectively. There were significant differences in vertical bone reconstruction and bone resorption between double-barrel flap and vertical distraction and between iliac crest and vertical distraction (*p* < 0.001). The study did not show significant differences in implant failure (*p* = 0.346). Radiotherapy did not affect vertical bone reconstruction (*p* = 0.125) or bone resorption (*p* = 0.237) but it showed higher implant failure in radiated patients (*p* = 0.015). The double-barrel flap and iliac crest graft showed better stability in the height of bone and less bone resorption and higher implant success rates compared with vertical distraction. Radiation therapy did not affect the vertical bone reconstruction but resulted in a higher implant failure.

## 1. Introduction

Mandibular defects derived from trauma, congenital malformations and tumor resection cause severe bone and soft tissue defects, with their consequent aesthetic and functional sequelae [1]. Segmental mandibulectomy leads to mandibular deviation, malocclusion, temporomandibular joint disorders and restriction to a soft diet [2]. From the aesthetic point of view, there is a retrusion of the lower third of the face, especially if the mandibulectomy includes the symphysis and mandibular body. In these cases, there is also severe ptosis of the lower lip. When the resection affects the mandibular body, there is clear facial asymmetry with collapse of the affected side [1]. This asymmetry is more evident if the condyle is included in the resection. Functionally, the most important sequelae are incompetence of the lower lip, salivary incontinence, severe difficulty in chewing and swallowing, and language disorders. On the one hand, the unreconstructed mandible tends to retrude and deviates toward the affected side. On the other hand, previous vertical movements are replaced by oblique or diagonal movements controlled by a single temporomandibular joint. The tongue is limited in mobility and strength, and proprioceptive sensitivity disorders lead to incoordination of mandibular movements. From a professional and social point of view, the patients are clearly diminished and, in many cases, totally prevented from carrying out these functions.

In the last forty years, due to the development of reconstructive techniques in the head and neck, there has been a significant improvement in the comprehensive treatment of these patients. Microsurgical techniques, virtual surgical planning (VSP), computer-aided design/computer-aided manufacturing (CAD/CAM), surgical navigation and advanced implantology have improved the aesthetic and functional results [3,4,5]. Free flaps are considered the treatment of choice in mandibular reconstruction for extensive bone defects [6]. The iliac crest flap, scapular flap and fibula flap are the main options for mandibular reconstruction. Since Hidalgo described the use of the fibula flap for mandibular reconstruction, this flap has become the flap of choice for the reconstruction of composite mandibular defects [7]. The fibula flap offers several advantages including: (1) great length of bone, being the only flap that allows the reconstruction of defects over 12 cm [8]; (2) medullary and periosteal blood supply that gives us the possibility of multiple osteotomies to recontour the new mandible [9]; (3) long vascular pedicle and constant geometry [10]; (4) possibility of implant placement and oral rehabilitation with implant-retained or implant-supported prosthesis [11]; (5) a large skin paddle for soft tissue reconstruction [8]; (6) bicortical bone, which is ideal for the primary stability of dental implants and long-term implant prosthetic rehabilitation [12,13]; (7) low donor site morbidity and (8) a two-team approach for the surgical procedure.

However, the fibula flap does not provide sufficient height of bone to restore the native height of the mandible. The vertical discrepancy between the fibula flap and native mandible results in: (1) shortening of the vertical dimension of the lower third of the face with the consequent aesthetic defects and (2) difficulty in implant placement and prosthetic rehabilitation that may cause implant overloading and endanger both the functional and aesthetic long-term results [6]. To solve this problem, we have three surgical options: (1) the use of a double-barrel fibula flap; (2) an iliac crest onlay graft; and (3) a vertical distraction osteogenesis of the fibula flap. All techniques provide sufficient height and width of bone for implant placement and prosthetic rehabilitation. The purpose of this study was to evaluate and compare the vertical reconstruction of the fibula flap with three different techniques and osseointegrated implants. The specific aims of this study were: (1) to compare the vertical bone reconstruction; (2) to compare perimplant bone resorption; (3) to compare the implant success rate; (4) to establish whether radiation therapy significantly affects vertical bone reconstruction and perimplant bone resorption; (5) to evaluate the association between radiotherapy and implant failure. The review of medical records and data collection and the subsequent analysis of the data collected is endorsed by the Hospital Ethics Committee at Gregorio Marañón General Hospital, Madrid, Spain.

## 2. Experimental Section

### 2.1. Material and Methods

To address the research purpose, the investigators designed and implemented a retrospective study during a 5-year period (2015–2019) with twenty-four patients (15 men and 9 women) with segmental mandibular defects reconstructed with a fibula free flap at Hospital General Universitario Gregorio Marañón (Madrid, Spain). The inclusion criteria were: (1) oncologic patients who were going to be treated with a segmental mandibulectomy; (2) patients with traumatological sequelae or deformities with a mandibular segmental defect; (3) patients with no previous history of radiotherapy or chemotherapy. In patients in whom oncologic surgery was planned with a segmental mandibulectomy and soft tissue resection, and in whom the reconstruction was performed with an osseocutaneous fibula flap, the reconstructive technique used was the double-barrel flap. In patients who did not require intraoral soft tissue reconstruction, the reconstruction was performed with an osseous fibula flap, and the patients were randomly assigned to the vertical reconstruction group with iliac crest graft or vertical distraction. The segmental defects originated as a result of tumor resection, traumatic or malformation defects. In 4 cases, the origin was a traumatic defect. One patient with right hemi-mandibular agenesis was referred to our Department for secondary mandibular reconstruction, and 19 patients underwent segmental mandibulectomy due to oncological disease. Thirteen oncologic patients were diagnosed with squamous cell carcinoma and 6 patients were diagnosed with ameloblastoma. Immediate reconstruction with a fibula free flap was accomplished in oncologic patients while a secondary reconstruction was performed in the other 5 cases. In all patients, an MRI checked the viability of the tibioperoneal vessels prior to surgery. In patients who required an intraoral soft tissue resection and a reconstruction with a fibula osseocutaneous flap, the perforators were marked through MRI and a skin paddle was outlined around the marked perforators. In patients who did not require soft tissue reconstruction, piercing marking was not necessary. In 8 patients, the previous mandibular height was reconstructed with a double-barrel fibula flap. In the other 16 patients a second surgery was performed to reconstruct the vertical dimension of the mandible and thus resolve the vertical discrepancy between the fibula and the remaining bone. In 8 patients, a vertical distraction of the fibula flap was performed, and in 8 patients, an onlay cortico-cancellous iliac crest bone graft was placed over the previous fibula flap.

Vertical reconstruction of the fibula was performed in 7 nonirradiated patients with an onlay iliac crest bone graft while one patient received radiotherapy. Under general anesthesia a two-team surgical approach was accomplished 6 months after prior surgery. A corticocancellous graft from the anterior superior iliac crest was harvested because of its thicker cortical layer. A cervical approach was performed to expose the fibula flap and remove the osteosynthesis material. In this way, the corticocancellous graft was isolated from the oral cavity. In 4 patients the graft was fixed using conventional titanium mesh adapted and fixed to the remaining fibula. A virtual surgical planning (VSP) was performed on 4 patients, and a custom-made titanium mesh (CAD/CAM) was designed. Six months later, an intraoral approach was planned, the titanium mesh was removed, and the Ticare^®^ dental implants were placed. Six months later, prosthetic rehabilitation was achieved (Figure 1 and Figure 2).

Vertical distraction was performed in a second surgical procedure on 8 patients. Four patients had not previously received radiotherapy and 4 patients were irradiated. Four months after reconstructive surgery, the osteosynthesis material was removed, and a Martin alveolar distraction device was placed. Under general anesthesia, the fibula was exposed through a vestibular incision, the osteosynthesis plates were removed and the distraction device was positioned on the vestibular bony surface preserving the lingual mucoperiosteal attachment. The osteotomies were marked, the distraction device was removed, and the osteotomies were performed with a sagittal saw. The device was repositioned and activated to a distance of 3 mm. After a 7-day healing period, the distraction was started at a rate of 1 mm per day. The distraction was stopped when it was proven radiologically that a sufficient amount of bone had been achieved to place the implants ensuring a correct implant/crown ratio. After a stabilization period of 12 weeks, the device was removed and dental implants were placed (Ticare^®^, Valladolid, Spain). Four months later, the prosthetic rehabilitation was accomplished with a fixed implant-supported prosthesis (Figure 3 and Figure 4).

In patients with a double-barrel flap, the fibula was osteotomized in two segments and a 2- to 3-cm segment of bone between the two layers of the double-barrel fibula was removed. The original mandibular contour was maintained using rigid fixation with a mandibular reconstruction plate between the remaining mandible and the lower layer of the fibula flap. Vertical reconstruction of the mandibular height was achieved through semirigid fixation with miniplates between the upper layer of the fibula and the upper border of the mandible. In a second surgical procedure, the miniplates were removed and Ticare^®^ dental implants were placed in the fibula for prosthetic rehabilitation. In patients who did not receive radiotherapy, the implants were placed 4 months later. In patients who had radiation therapy, implants were placed 1 year after irradiation was finished (Figure 5 and Figure 6). Functional parameters were assessed after prosthetic rehabilitation and the patients reported an unrestricted diet, normal speech, and good oral competence.

Once the vertical dimension of the mandible had been reconstructed, dental implants (Ticare^®^, Valladolid, Spain) were placed in all patients and were subsequently rehabilitated with a fixed implant-supported prosthesis with the aim of achieving a comprehensive reconstruction, both aesthetic and functional. In the postoperative follow-up, the increase in mandibular height achieved with the different reconstructive techniques, as well as the peri-implant bone resorption, the implant success rate and the effects of radiotherapy were evaluated. The measurement of the vertical reconstruction with the different techniques was performed in all patients before the placement of the dental implants and in the anatomical area where the placement of the implants was planned. The follow-up period was from 1 year 4 months to 4 years 11 months (average 3 years 8 months).

### 2.2. Statistical Analysis

Quantitative values were expressed as mean ± standard error of mean (S.E.M), while qualitative variables were expressed as frequencies and percentages. Kruskal-Wallis and Mann-Whitney’s tests were used to compare differences between groups of quantitave variables. Qualitative variables were compared using Chi-squared test. The statistical analysis was performed using the software SPSS 25.0. A two-tail *p* value < 0.05 was considered statistically significant.

## 3. Results

Twenty-four patients with segmental mandibular defects were reconstructed with a free fibula flap. In 19 patients the origin was oncological, 13 squamous cell carcinoma and 6 ameloblastomas. Four patients were reconstructed due traumatological sequelae and 1 patient due to right hemifacial microsomia. In 8 patients (33.3%), the previous mandibular height was immediately reconstructed with a double-barrel flap (Table 1). In 16 patients, the vertical reconstruction of the mandible was performed in a deferred surgery (66.6%). Of these 16 patients, in 8 patients, a corticocancellous iliac crest graft was performed (Table 2), and in another 8 patients, a vertical distraction of the fibula was accomplished (Table 3).

The smallest mandibular segmental defect was 6.3 cm, and the largest defect was 16.4 cm, with an average of 10.2 cm. Eleven flaps were bone flaps without skin paddles (45.9%), and thirteen flaps were osseocutaneous fibula flaps (54.1%). One flap required surgical revision for partial thrombosis of the venous anastomosis.

Patients reconstructed with the double barrel fibula flap presented an average age of 56.6 ± 4.7 years. 5 patients were men (62.5%) and 3 patients were women (37.5%). 3 patients did not receive radiation therapy (37.5%) and 5 patients were irradiated (62.5%). Vertical reconstruction was 18.5 ± 0.5 mm and bone resorption were 1.23 ± 0.09 mm (Table 1). A total of 28 implants were placed with an average of 3.5 ± 0.26 implants per patient. No major complications occurred, except intraoral exposure of osteosynthesis material in one patient who was treated conservatively. This technique allowed the reconstruction of the original height of the mandible. A total of 28 implants (Ticare^®^, Valladolid, Spain) were placed. Five patients were irradiated to 60 Gy, and 1 year after finishing the radiotherapy, 19 implants (67.8%) were placed. In the three no irradiated patient, 9 implants (32.2%) were placed 4 months after surgery. All no irradiated implants were correctly osseointegrated. One of the 19 irradiated implants failed during the osseointegration period (5.2%) and 18 implants osseointegrated correctly (94.7%). All patients were rehabilitated with a fixed implant-borne prosthesis. The total osseointegration rate of the implants in the double-barrel flap was 96.4%. One year after prosthetic rehabilitation, mesial and distal peri-implant bone resorption was evaluated. The average bone resorption was 1.23 mm. While in irradiated implants, the resorption was 1.4 mm, 1.3 mm, 1.5 mm, 1.4 mm and 1.5 mm, respectively, in no irradiated implants the resorption was no significant (0.9 mm, 1 mm and 0.9 mm) (Table 1).

Patients reconstructed with the iliac crest graft had an average age of 48.3 ± 6.4 years. 5 patients were men (62.5%) and 3 patients were women (37.5%). 7 patients did not receive radiation therapy (87.5%) and 1 patient was irradiated (12.5%). Vertical reconstruction was 17.75 ± 0.6 mm and bone resorption were 1.43 ± 0.04 mm (Table 2). A total of 38 implants were placed with an average of 4.75 ± 0.4 implants per patient. One patient received radiotherapy and the surgery was deferred 12 months. Seven patients did not receive radiotheraphy and the vertical reconstructive surgery was performed 4 months later. All patients underwent surgery in a 2-team surgical procedure with a cervical approach and an anterosuperior iliac crest approach. Through the cervical approach, the fibula was exposed, the osteosynthesis material was removed and the graft was placed. The fixation of the graft was accomplished with titanium mesh. In 4 patients, the mesh was hand-shaped and in 4 patients a VSP was performed before surgery for the design and printing with CAD/CAM techniques. None of the grafts communicated with the oral cavity. Six months later, once the ossification of the graft was checked by panoramic radiograph and CT scan, an intraoral approach was accomplished, the titanium meshes were removed and dental implants were placed (Ticare^®^, Valladolid, Spain). A total of 38 implants were placed, with an osseointegration success rate of 94.7%. Two implants were lost during the osseointegration period (5.3%). All patients were rehabilitated with a fixed implant prosthesis.

Patients reconstructed with vertical distraction presented an average age of 51.2 ± 6.8 years. 5 patients were men (62.5%) and 3 patients were women (37.5%). 4 patients did not receive radiotherapy (50%) and 4 patients were irradiated (50%). Vertical reconstruction was 12.5 ± 0.78 mm and bone resorption were 2.31 ± 0.12 mm (Table 3). A total of 32 implants were placed with an average of 4 ± 0.3 implants per patient.

All patients underwent an intraoral approach for distractor placement. Four patients did not receive radiation therapy and did not present with any complications. In these four patients, vertical distraction was performed 4 months after the first surgery. In the four patients who received radiotherapy (60 Gy), the distraction was performed one year after the completion of radiotherapy. The patients presented with intraoral distractor exposure that required antibiotic therapy and enteral nutrition. In these patients, the vertical increase in bone height was 8 mm, 11 mm, 13 mm and 12 mm, respectively. A total of 32 dental implants were placed in these 8 patients. Fourteen implants (43.75%) did not receive radiotherapy, and 18 implants (56.25%) received the maximum dose of radiotherapy. In the patients who received radiotherapy, vertical bone resorption was identified. Eighteen implants could be placed with bone remodeling, after removal of the distractor. All no irradiated implants were rehabilitated with a fixed implant prosthesis. Of the 18 irradiated implants, four implants failed during the osseointegration period. However, the remaining 14 implants were successfully rehabilitated. The implant success rate in patients reconstructed with vertical distraction was 87.5% with a failure rate of 12.5%.

### 3.1. Vertical Bone Reconstruction

Kruskal Wallis’ statistical analysis shows significant differences in bone growth between the three techniques used (*p* < 0.001) (Table 4). The Mann-Whitney analysis determines if there are significant differences between the techniques peer-to-peer comparison. According to this technique, there are no significant differences between the double barrel technique and the iliac crest technique (*p* = 0.485). However, there are significant differences between vertical distraction and double barrel (*p* = 0.001) and between vertical distraction and iliac crest (*p* = 0.001) (Figure 7).

### 3.2. Bone Resorption

The statistical analysis shows significant differences in bone resorption between the three techniques used (*p* < 0.001) (Table 4). The Mann-Whitney analysis determines that there are no significant differences between the double barrel technique and the iliac crest technique (*p* = 0.104). However, there are significant differences between vertical distraction and double barrel (*p* = 0.001) and between vertical distraction and iliac crest (*p* = 0.001) (Figure 8).

### 3.3. Implant Success Rate

The implant success rate was 96.4% (1 failure) in double barrel technique, 94.7% (2 failures) in vertical reconstruction with iliac crest graft and 87.5% (4 failures) in vertical distraction osteogenesis. Statistical Chi-square analysis determines whether there is a significantly different distribution of failures among the three reconstruction groups. The study does not show significant differences in implant failures between the three techniques used (*p* = 0.346) (Table 5).

### 3.4. Effects of Radiation Therapy in Bone Reconstruction Bone Resorption

Mann-Whitney’s statistical analysis compares the variables of bone height (mm) and bone resorption (mm) between irradiated and non-irradiated patient groups. The study shows no significant difference between the two groups and therefore radiation therapy does not affect vertical bone reconstruction (*p* = 0.125) or bone resorption (*p* = 0.237) (Table 6).

### 3.5. Association between Radiotherapy and Implant Failure

57 implants were placed in non-irradiated patients with only 1 implant failure (98.2% success rate). 41 implants were placed in irradiated patients with 6 implant failure (85.3%). Statistical Chi-square analysis showed significant difference between the two groups (*p* = 0.015) (Table 7). All patients were rehabilitated with a fixed implant-supported prosthesis, recovering aesthetics and function with intelligible speech and an unrestricted diet.

## 4. Discussion

The main disadvantage of the fibula flap is the low height of bone to reconstruct the height of the previous mandible [2]. If the defect is extensive and affects the area between the mandibular angles, there is a decrease in the vertical dimension resulting in a shortening of the face [14]. Despite this, it can be rehabilitated with osseointegrated implants, bearing in mind that the crown/implant ratio must be correct in order not to overload the implants occlusally. Another option would be to perform a vertical reconstruction in the symphysis and mandibular body to increase the height of the lower third of the face [15]. In cases of mandibular segmental defects with a remaining mandible and occlusion on one or both sides of the defect, there is a vertical discrepancy between the fibula and remnant mandible, which makes correct prosthetic rehabilitation with implants impossible because the prosthesis would be extremely high and occlusally inadvisable [16]. To solve this problem, we can reconstruct the vertical defect using three techniques: double-barrel flap, vertical bone distraction and iliac crest onlay graft on the previous fibula.

The aims of this study were to compare the vertical bone reconstruction, the peri-implant bone resorption and the implant success rate between the three techniques, to establish whether radiation therapy significantly affects vertical bone reconstruction and peri-implant bone resorption and to evaluate the association between radiotherapy and implant failure. Our study revealed that vertical bone augmentation is higher in patients reconstructed with the double-barrel technique and the iliac crest graft. The double-barrel flap depends on the height of the fibula of the patient, determines the height during surgery, increases the bone height by twice and is more predictable. The iliac crest graft develops postoperative absorption, and that is the main reason why the investigators initially overcorrect the amount of bone over the fibula. In these patients, vertical reconstruction is therefore less predictable than in vertical distraction or double-barrel flap. Surprisingly, the bone gain is similar to the double-barrel and higher than in distraction, probably due to the absence of postoperative radiotherapy. The vertical reconstruction with the distraction technique showed that the bone gain was lower in patients who underwent radiotherapy, while non-irradiated patients developed a stable bone gain between 13 and 15 mm. In this context, although distraction was stopped when there was radiologically evidence of sufficient vertical bone formation for a correct implant/crown ratio, our recommendation would be to overcorrect the distraction, specially, in patients with prior radiotherapy. Our study reveals that the bone resorption is lower in reconstructed patients with double-barrel flap and iliac crest graft. Although no significant shortening of the distraction appeared after the end of distraction, the vertical distraction of the fibula results in less vertical bone increase and more peri-implant bone resorption, due to the effect of radiation therapy on distracted bone. The limitation of this study is the sample size. Therefore, although the survival of implants is higher in double-barrel reconstruction and iliac crest graft, statistical analysis shows no significant differences between the three groups. As for the role of radiotherapy, the study shows no significant differences in vertical bone augmentation or bone resorption between radiated and non-irradiated patients. However, irradiated patients present a higher rate of implant loss compared to non-irradiated patients.

To date, this is the first study that compares the vertical ridge augmentation of the fibula flap with double-barrel flap, iliac crest graft and vertical distraction osteogenesis. Yue He [2] reconstructed 7 patients with the double-barrel fibula flap with a reconstruction height of 3.0–3.8 cm. Three patients were irradiated and only 1 irradiated patient was rehabilitated with implants. Furthermore, no data about bone resorption was described. Shen [17] reported 45 double-barrel flaps with implant rehabilitation in 11 patients with good functional and esthetic results, although no implant success rate or bone resorption were described. Ferreti [18] and Sethi [19] described vertical reconstruction with iliac crest graft in atrophic mandible but they do not report segmental mandibulectomy or reconstruction with fibula flap. Lizio [20] reported 5 cases of vertical distraction of the fibula flap in mandibular reconstruction with a mean vertical bone gain of 14 mm and a mean peri-implant bone resorption of 2.5 mm. and reported numerous complications of both hard and soft tissues, illustrating the critical importance of selection of patients. Another limitation of our study is that it does not compare the ratio of the fibula width and the ratio of healthy mandible. Further studies should be performed to compare the relative values considering that the size varies from patient to patient.

At this point and, considering that the three techniques are appropriate for vertical reconstruction of the fibula, the main question is: what are the benefits and in which cases should we use each technique?

The double-barrel fibula flap is the ideal technique in order to reconstruct the mandibular height and to solve the problem of vertical discrepancy. It has the following advantages: (1) a great length of bone that allows us to reconstruct defects in the double-barrel between 8 cm and 10 cm; (2) no delayed surgery compared with the vertical distraction and onlay grafts [18]; (3) great suitability for dental implants and early prosthetic rehabilitation [2]; (4) high vascularity allows us to perform osteotomies to shape the double-barrel flap without vascular compromise of the flap; (5) possibility of using an osteocutaneous flap for the reconstruction of composite defects where an associated soft tissue defect needs to be reconstructed; (6) low donor site morbidity; (7) it is the better technique in patients who are going to receive radiotherapy due to the high vascularity of the flap with minimal bone resorption; and (8) minimal peri-implant bone resorption. The disadvantage of this technique is that it is not advisable to reconstruct defects greater than 10 cm because the length required is almost 24 cm, which can lead to higher donor site morbidity. It is also possible to vertically reconstruct only a part of the segmental defect necessary for implant placement [21,22,23]. Therefore, in oncological patients, especially those who are going to be irradiated, and in patients with mandibular segmental defects of up to 10 cm, this is the surgical technique of choice. It allows immediate reconstruction and fast prosthetic rehabilitation with a low rate of peri-implant bone resorption and high implant success rate, even in irradiated patients.

A cortico-cancellous iliac crest bone graft allows the vertical dimension of the fibula to be reconstructed in a deferred surgery with low bone resorption and high implant success rate. It provides an adequate volume of highly cellular autologous bone [18]. It has the advantage of providing cortico-cancellous bone with good irrigation. However, it has several disadvantages: (1) it requires a second surgical procedure; (2) it is necessary to overcorrect approximately 25% of its height to compensate its partial resorption and carry out the subsequent remodeling [19]; (3) morbidity is derived from the approach to the iliac crest; (4) there is partial vertical resorption of the graft during ossification; (5) there is a possibility of exposure of the titanium mesh and bone graft, especially in irradiated patients; and (6) there is need to wait 6 months for ossification before placing the implants, making the prosthetic rehabilitation longer. This technique is indicated in patients who are not going to receive radiotherapy and in patients with extensive defects at the level of the symphysis and mandibular body. When this technique is accomplished, it is important to perform a cervical approach for the placement of the graft and avoid communication with the oral cavity. Implant success rate, and peri-implant bone resorption are similar to the one found in the double-barrel flap, but the higher number of surgeries, higher morbidity and complication rate, and slower prosthetic rehabilitation make the onlay graft a second-choice technique whenever a double-barrel fibula flap can be used.

Distraction osteogenesis can be used for vertical fibula reconstruction and has been widely described in the literature [6,24]. It has the advantage of being able to control the distraction vector, direct bone gain and soft tissue augmentation, but there are several disadvantages, including: (1) it requires a second surgical procedure and the placement of a vertical distractor, normally through an intra-oral approach; (2) it requires a period of 15–18 weeks between the latency, distraction and consolidation periods; (3) it requires a third surgical procedure for the removal of the distractor and placement of the implants; (4) there is a possibility of bone and distractor exposure and dehiscence in irradiated patients; (5) bone resorption of the vertical bone is obtained; (6) peri-implant bone resorption and implant failure is greater than in other techniques [24]; and (7) slower prosthetic rehabilitation. Vertical distraction is a surgical technique that we recommend for the reconstruction of small extensive and vertical defects in nonirradiated patients. Complications in irradiated and nonirradiated patients are multiple, including distractor exposure, distraction failure and increased bone and peri-implant resorption.

## 5. Conclusions

Vertical ridge augmentation is higher, and the bone resorption is lower in reconstructed patients with double-barrel flap and iliac crest graft. The vertical distraction of the fibula results in less vertical bone increase and more peri-implant bone resorption. There are no significant differences in implant success rate between techniques, but irradiated patients present a higher rate of implant loss compared to non-irradiated patients. Finally, it is important to emphasize that it is necessary to acknowledge all these surgical techniques, in order to individualize the vertical reconstruction of the fibula flap according to the pathology, defect and the characteristics of patients. The combination of these techniques together with advanced implantology allows the comprehensive rehabilitation of the patients, providing aesthetic and functional results that return quality of life to patients.

## Figures and Tables

**Figure 1 jcm-10-00101-f001:**
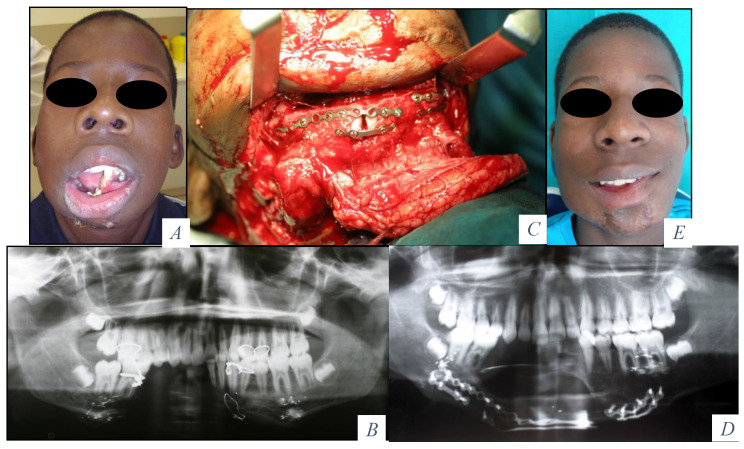
(**A**) Labial incompetence, salivary incontinence, impossibility to accomplish a normal diet and speech. (**B**) Segmental bone defect at the symphysis and right mandibular body. (**C**) Fibula osteocutaneous flap for composite reconstruction. Semirigid fixation. (**D**) Panoramic radiograph after mandibular reconstruction. (**E**) Recovery of mandibular symmetry, oral opening and oral competence.

**Figure 2 jcm-10-00101-f002:**
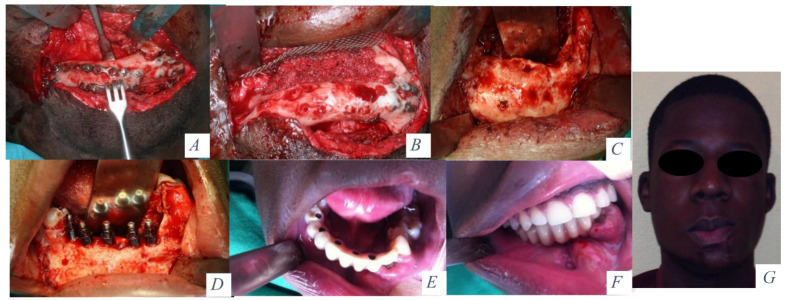
(**A**) Cervical approach and removal of the osteosynthesis material. (**B**) Corticocancellous iliac crest bone graft and titanium mesh. (**C**) Mandibular height and width gained with the graft 6 months later. (**D**) Placement of 5 implants (Ticare) in the neomandible. (**E**) Prosthetic rehabilitation with a fixed implant supported prosthesis. (**F**) Final occlusion. (**G**) Final aesthetic result.

**Figure 3 jcm-10-00101-f003:**
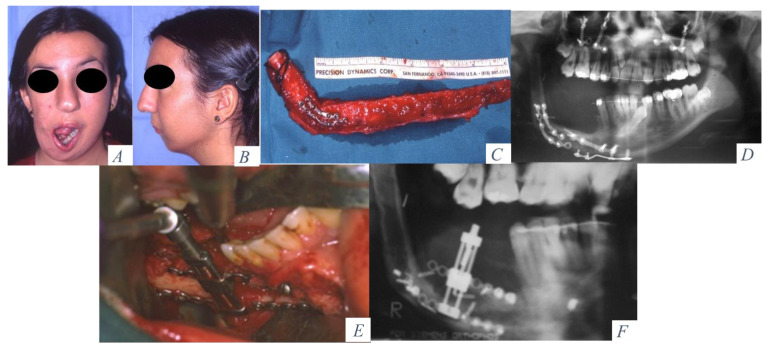
(**A**) Hemimandibular agenesis with facial asymmetry and mandibular deviation. (**B**) Retrusion of the lower facial third. (**C**) Mandibular reconstruction with an osseous fibula flap. (**D**) Panoramic radiograph with vertical discrepancy between the remaining mandible and the fibula flap. Le Fort I osteotomy for occlusal compensation. (**E**) Alveolar distractor placed in the fibula. (**F**) Panoramic radiograph at the beginning of the distraction procedure.

**Figure 4 jcm-10-00101-f004:**
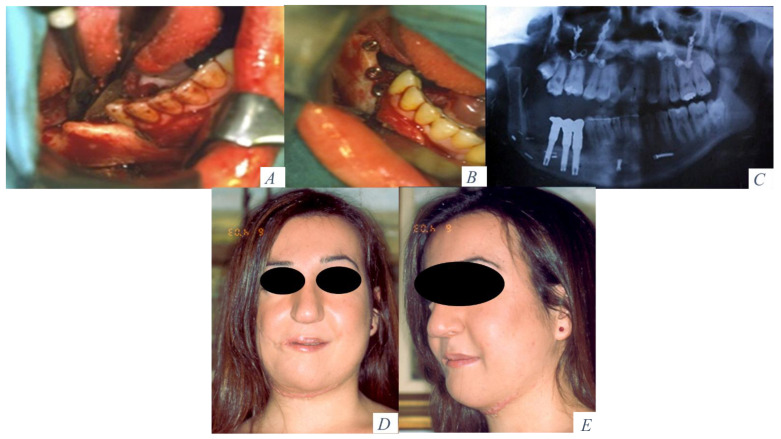
(**A**) Height and width of bone obtained after alveolar distraction. (**B**) Placement of 3 dental implants in the fibula flap. (**C**) Prosthetic rehabilitation with a fixed implant-supported prosthesis. (**D**) Facial symmetry and aesthetic result after mandibular reconstruction and oral rehabilitation. (**E**) Projection of the lower third face.

**Figure 5 jcm-10-00101-f005:**
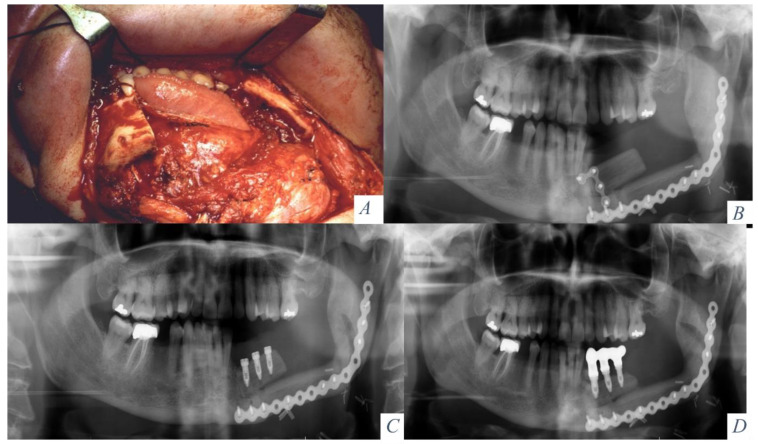
(**A**) Squamous cell carcinoma of the floor of the mouth with mandibular invasion. Tumor resection with segmental mandibulectomy and clear margins. (**B**) Immediate reconstruction with a double-barrel osseocutaneous free flap. Rigid fixation between the lower layer of the fibula and the remaining mandible. Semirigid fixation between the upper layer and the mandible. (**C**) Removal of part of the osteosynthesis material and placement of three implants (Ticare). (**D**) Prosthetic rehabilitation with a fixed implant supported prosthesis.

**Figure 6 jcm-10-00101-f006:**
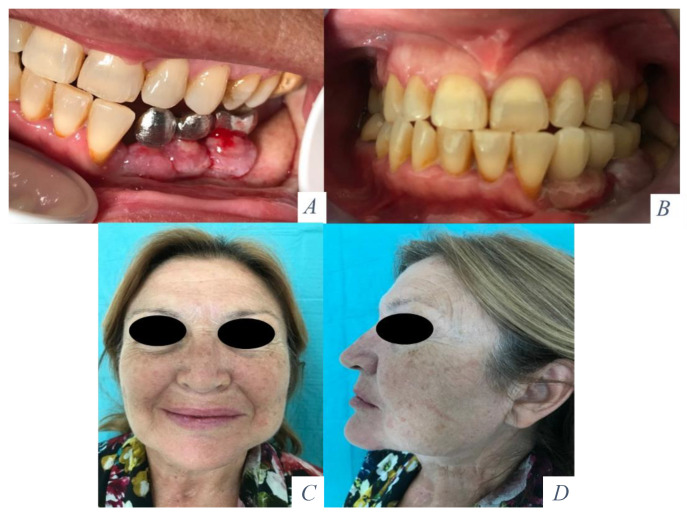
(**A**) Metal framework evaluated intraorally. (**B**) Functional rehabilitation with a ceramic fixed implant supported prosthesis. (**C**) Aesthetic result with mandibular symmetry. (**D**) Aesthetic profile with a good projection of the lower facial third.

**Figure 7 jcm-10-00101-f007:**
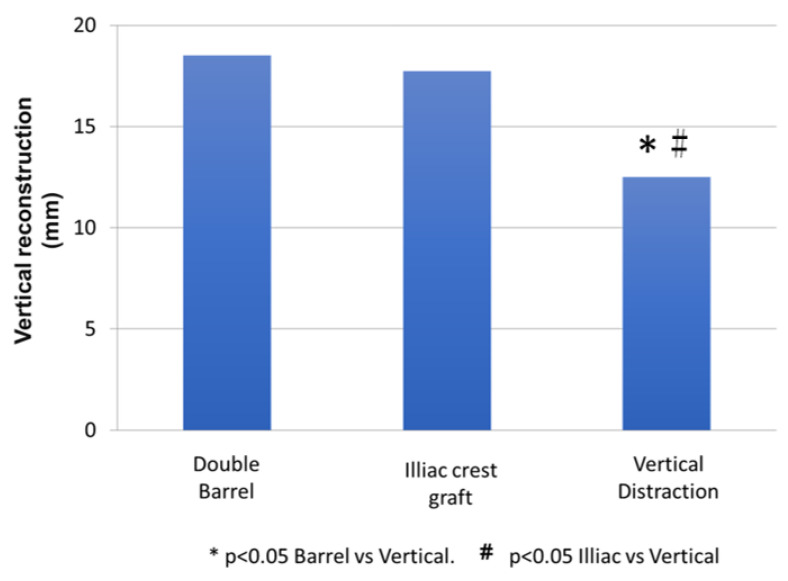
Mann-Whitney analysis for vertical bone reconstruction.

**Figure 8 jcm-10-00101-f008:**
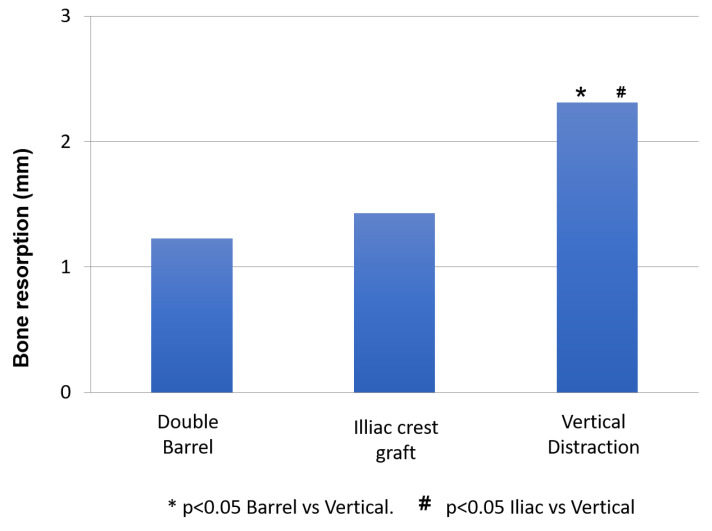
Mann-Whitney analysis for bone resorption.

**Table 1 jcm-10-00101-t001:** Vertical reconstruction with double-barrel free flap.

	Gender/Age	Diagnosis	Length of Defect (cm)	Vertical Reconstruction (mm)	Number of Implants/(Failure)	Bone Resorption (mm)	Radiotherapy
Double-barrel	F/64	Squamous cell carcinoma	8.7	18	3	1.4	60 Gy
Double-barrel	M/68	Squamous cell carcinoma	10.2	18	5 (1 failure)	1.3	60 Gy
Double-barrel	M/72	Squamous cell carcinoma	9.6	19	4	1.4	60 Gy
Double-barrel	F/62	Squamous cell carcinoma	8.3	17	3	1.5	60 GY
Double-barrel	F/47	Ameloblastoma	8.2	17	3	0.9	No
Double-barrel	M/63	Squamous cell carcinoma	9.3	18	4	1.5	60 Gy
Double-barrel	M/34	Traumatic injury	7.2	21	3	1.0	No
Double-barrel	M/43	Traumatic Injury	6.8	20	3	0.9	No
Rate	56.6 ± 4.7			18.5 ± 0.5 mm	28 (1 failure): 96.4%	1.23 ± 0.9 mm	

**Table 2 jcm-10-00101-t002:** Vertical reconstruction with iliac crest bone graft.

	Gender/Age	Diagnosis	Length of Defect (cm)	Vertical Reconstruction (mm)	Number of Implants/(Failure)	Bone Resorption (mm)	Radiotherapy
Iliac crest	M/13	Traumatic injury	9.2	18	5	1.4	No
Iliac crest	M/44	Ameloblastoma	9.5	19	4	1.5	No
Iliac crest	F/59	Squamous cell carcinoma	6.8	17	4	1.5	No
Iliac crest	F/61	Squamous cell carcinoma	8.5	16	4	1.6	No
Iliac crest	M/59	Traumatic injury	9.1	18	6	1.2	No
Iliac crest	M/43	Ameloblastoma	10.6	19	7 (1 failure)	1.5	No
Iliac crest	F/37	Ameloblastoma	8.4	20	4	1.4	No
Iliac crest	M/71	Squamous cell carcinoma	8.1	15	4 (1 failure)	1.4	60 Gy
Rate	48.3 ± 6.4			17.75 ± 0.6 mm	38 (2 failures): 94.7%	1.43 ± 0.04 mm	

**Table 3 jcm-10-00101-t003:** Vertical reconstruction with vertical distraction osteogenesis.

	Gender/Age	Diagnosis	Length of Defect (cm)	Vertical Reconstruction (mm)	Number of Implants/Failure	Bone Resorption (mm)	Radiotherapy
Vertical distraction	F/16	Hemifacial microsomia	16.4	13	3	2.0	No
Vertical distraction	M/32	Ameloblastoma	13.2	15	5	1.9	No
Vertical distraction	M/62	Squamous cell carcinoma	13.7	8	5 (1 failure)	2.6	60 Gy
Vertical distraction	F/59	Squamous cell carcinoma	6.3	14	3	2.1	No
Vertical distraction	F/41	Ameloblastoma	8.3	14	3	2.0	No
Vertical distraction	M/62	Squamous cell carcinoma	9.5	11	5 (1 failure)	2.6	60
Vertical distraction	M/67	Squamous cell carcinoma	8.6	13 mm	4 (1 failure)	1.9	60 Gy
Vertical distraction	M/71	Squamous cell carcinoma	9.1	12 mm	4 (1 failure)	2.2	60 Gy
Rate	51.2 ± 6.8			12.5 ± 0.78 mm	32 (4 failure): 87.5%	2.31 ± 0.12 mm	

**Table 4 jcm-10-00101-t004:** Kruskal Wallis’ statistical analysis for vertical reconstruction and bone resorption.

Kruskall Wallis	Double Barrel	Iliac Crest	Vertical Distraction	*p*
Vertical reconstruction (mm)	18.50 ± 0.5	17.75 ± 0.6	12.50 ± 0.78	<0.001
Bone resorption (mm)	1.2375 ± 0.09	1.4375 ± 0.042	2.3125 ± 0.12	<0.001

**Table 5 jcm-10-00101-t005:** Implant failure between the different techniques.

			Implant Failure		*p* Value
			**NO** ***n* (%)**	**YES** ***n* (%)**	0.346
TECHNIQUE	Double barrel	Rate	27 (29.7%)	1 (14.3%)	
	Iliac crest graft	Rate	36 (39.6%)	2 (28.6%)	
	Vertical distraction	Rate	28 (30.8%)	4 (57.1%)	

**Table 6 jcm-10-00101-t006:** Effects of radiotherapy in vertical reconstruction and bone resorption.

	Radiotherapy	No Radiotherapy	*p*
Vertical Reconstruction (mm)	14.90 ± 1.18	17.21 ± 0.67	0.125
Bone resorption (mm)	1.9 ± 0.19	1.49 ± 0.10	0.237

**Table 7 jcm-10-00101-t007:** Association between radiotherapy and implant failure.

			Implant Failure		*p* Value
			**NO** ***n* (%)**	**YES** ***n* (%)**	0.015
Radiotherapy	NO	Rate	56 (61.5%)	1 (14.3%)	
	YES	Rate	35 (38.5%)	6 (85.7%)	

## Data Availability

The data presented in this study are available on request from the corresponding author. The data are not publicly available due to data protection regulations.

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
