# Peer review of "Vertical Ridge Augmentation of Fibula Flap in Mandibular Reconstruction: A Comparison between Vertical Distraction, Double-Barrel Flap and Iliac Crest Graft"

_jcm, 2020, doi:10.3390/jcm10010101_

Round 1
Reviewer 1 Report
Comments on ”Vertical ridge augmentation of fibula flap in mandibular reconstruction: ---“
The manuscript is interesting reading.
However, as a scientific article, for the different parts (Introduction, Materials and Methods, Results and Discussion) the headings are given but the content is mixed between the sections, and the same information is often given in more than one section. For example in the discussion, to a great part, information is given that belongs to the introduction. There is much double information, and the results are given both in the text and in tables. These circumstances make the manuscript long, one gets tired to have the information repeated and it is less easy to catch the results. You list many things that you want to study, but it is not easy to find the results.
The part with the case presentations doesn´t fit in, I suggest that you publish that in another and separate article.
The discussion is less of discussion of the results but more another introduction. I miss a clear conclusion at the end of the discussion.
From the circumstances presented above, I would suggest
-that you collect the information about the techniques to the introduction,
-that you in Material and Method better describe the inclusion criteria of the patients and how the inclusion was made (how did you manage to get eight patients to each surgical technic?), and here you can describe how the surgery was performed but not elsewhere.
-under results remove the case reports. Try to describe your results according to your aims.
-under discussion, discuss your results, why you got that particular result, and add a clear conclusion.
Reviewer 2 Report
・Delete unnecessary information that is not relevant to the discussion, as the representation is verbose and the volume is too large throughout the treatise.
・Some figures are missing and cannot be seen. Is it a system problem?
・Please trim the images presented for comparison.
・It is necessary to specify the measurement location of the length of vertical reconstruction.
・In the text, the actual measured numbers of vertical reconstruction are compared. However, considering that the size varies from patient to patient, relative values such as the ratio of fibula width and the ratio of healthy mandible should be compared.
・The double barrel determines the height during surgery, the distraction determines the height during distraction, and the iliac bone graft has postoperative absorption to determine the height.
・It seems that only the iliac bone graft has an uncertain factor in the vertical length. Is there any change such as shortening the distraction after the distraction ends?
・Please specify the criteria for stopping distraction in material and methods
・In material and methods, specify when to evaluate the length of vertical reconstruction for each method.
・Since the height is artificially determined in Distraction, it is meaningless to make a significant difference with respect to the height. However, it is meaningful to make a statistically significant difference if it is lowered for some reason aiming at the same height by all methods. In that case, please specify the reason.
Author Response
Please see the attachment

This manuscript is a resubmission of an earlier submission. The following is a list of the peer review reports and author responses from that submission.